# OpenReview forum: "Court Simulation with Multifaceted Agent Judgment for LLM Jailbreak Evaluation"
_ICLR.cc/2026/Conference — ICLR 2026 Conference Withdrawn Submission_

### Official Review · Reviewer_BYcp · 2025-10-30

**Soundness:** 2
**Presentation:** 3
**Contribution:** 2
**Rating:** 4
**Confidence:** 4

**Summary:**

The paper proposes CLEAR, a court-inspired, multi-agent protocol for LLM jailbreak evaluation that first aggregates four kinds of structured evidence and then issues a first-instance judgment without exposing the judge to the raw model output. Low-confidence cases trigger a public debate/appeal between pro- and con-agents, adjudicated by a second judge. Experiments on MI-JB, AB-JB, and the RobustEval (proposed by this paper) report Acc/F1/Recall, include ablations and confidence-threshold sensitivity, and claim stronger alignment with human judgments, especially on borderline cases.

**Strengths:**

[Strength 1]: The paper propose a clear, court-style multi-agent evaluation pipeline with strong interpretability (structured evidence, reasoned judgments).
[Strength 2]: The paper explicitly targets borderline/ambiguous scenarios, addressing two common pitfalls of prior protocols: incomplete definitions and unsystematic processes

**Weaknesses:**

[Weakness 1]: The overall idea is close to JAILJUDGE[1] (jailbreak judging with multi-agent system). The paper does not analyze the differences in depth and does not include JAILJUDGE as a baseline, limiting comparative credibility.
[Weakness2]: Dataset scale is small, and widely used jailbreak-evaluation sets (e.g., JBB Behaviors) are not covered, reducing statistical power and external validity.

[1] Liu, Fan, et al. "Jailjudge: A comprehensive jailbreak judge benchmark with multi-agent enhanced explanation evaluation framework." arXiv preprint arXiv:2410.12855 (2024).

**Questions:**

see weakness

---

### Official Review · Reviewer_Uef5 · 2025-10-31

**Soundness:** 2
**Presentation:** 3
**Contribution:** 2
**Rating:** 2
**Confidence:** 3

**Summary:**

This paper proposes CLEAR, a multi-agent, two-stage protocol for evaluating LLM jailbreaks. Stage 1 runs four specialized analyst agents on each prompt–response pair: a Retrieval agent (policy violations), a Knowledge Judge (intent fulfillment), a Harm Scorer (severity, 0–1), and a Behavior Tracer (stance shifts). An Extractor pre-filters disclaimers/verbosity before intent/harm scoring and the evidence is pooled. A first-instance judge then issues a label and confidence under a four-level taxonomy {Full Failure, Partial Failure, Partial Success, Full Success}. Low-confidence cases trigger Stage 2, a second-instance trial: prosecution/defense agents debate and a review judge returns a final verdict. Algorithm 1 formalizes the pipeline.

Empirically, CLEAR is evaluated using GPT-4o / GPT-4o-mini as the backbone evaluator on three datasets: MI-JB (97 items from MaliciousInstruct), AB-JB (100 prompts derived from AdvBench), and RobustEval (40 curated borderline cases). CLEAR achieves near-perfect scores on MI-JB, AB-JB and large gains on RobustEval vs. GPT-labeling, GPT-ReCheck, and a fine-grained baseline. Ablations indicate the multi-view evidence, extractor, and debate each contribute, and a sensitivity study explores the confidence threshold $\delta$.

**Strengths:**

1. Principled escalation via confidence-triggered second-instance debates reduces over-confident mislabels on ambiguous cases (ablation supports benefit of each component).

**Weaknesses:**

1. The idea of structured, court-inspired adjudication with multi-agent evidence and a confidence-gated debate stage is like a design containing lots of existing things. In other words, conceptual novelty over prior multi-judge/debate evaluators is very limited.
2. Empirical gains are shown but shown on relatively small, curated sets. MI-JB (97) and AB-JB (100) are small; RobustEval uses only 40 items for main results.
3. The policy repository S isn’t clearly described (which providers, which policies, languages?)
4. Stripping disclaimers/verbosity can remove refusal cues and make borderline refusals look like harmful answers. Please show flips caused only by the extractor with examples?
5. No error bars, no significance tests

**Questions:**

1. Who produced/verified ground truth for the binary mapping? Inter-annotator agreement? Any disagreements for PF vs. PS that the debate stage resolves?

---

### Official Review · Reviewer_xstU · 2025-11-01

**Soundness:** 2
**Presentation:** 2
**Contribution:** 2
**Rating:** 2
**Confidence:** 5

**Summary:**

This paper introduces CLEAR (Court Simulation with Multifaceted Agent Judgment), a highly complex multi-agent framework that addresses the prevalent ambiguity and inconsistency in Large Language Model (LLM) jailbreak evaluation. The framework innovatively models the evaluation process as a court trial, employing "multi-view evidence" extraction across four dimensions (policy violation, intent, harm score, and behavior pattern) and a two-tiered "debate" mechanism. This approach significantly improves accuracy, particularly for ambiguous, borderline cases (validated on the authors' new RobustEval dataset). While its core strength lies in systematically decomposing complex judgments and enhancing weaker models' evaluation capacity, the framework suffers from a critical weakness: its extreme complexity and prohibitive cost due to numerous LLM calls, rendering it largely impractical for real-world deployment beyond expensive dataset annotation. Furthermore, its efficacy is fundamentally constrained by the inherent capabilities and biases of its GPT-4o backbone, failing to solve the "who watches the watchmen" problem, and it carries the risk of losing critical context during the Extractor Agent's preprocessing step.

**Strengths:**

(1) The paper introduces the highly complex "court simulation" multi-agent framework (CLEAR) to evaluate jailbreak behavior. This innovative approach addresses the subjectivity and opacity of existing evaluation methods, offering a fresh, structured perspective.

(2) It decomposes complex jailbreak judgments into four critical evidence dimensions (e.g., policy violation, behavioral pattern tracking), particularly excelling at capturing ambiguous LLM behaviors like "refusal followed by compliance," which enhances the comprehensiveness and objectivity of the assessment.

(3) It decomposes complex jailbreak judgments into four critical evidence dimensions (e.g., policy violation, behavioral pattern tracking), particularly excelling at capturing ambiguous LLM behaviors like "refusal followed by compliance," which enhances the comprehensiveness and objectivity of the assessment.

**Weaknesses:**

(1)The framework is extremely complex, requiring numerous LLM calls for a single complete evaluation, leading to prohibitively high computational costs that severely limit its feasibility for practical applications like real-time monitoring.

(2)It is essentially a form of advanced prompt engineering and Chain-of-Thought distillation based on GPT-4o. The evaluation ceiling is ultimately constrained by the backbone model's inherent knowledge and biases, failing to resolve the fundamental limitations of the LLM-as-Judge paradigm.

(3)The critical Extractor Agent preprocessing step carries the risk of mistakenly removing essential contextual information (such as refusal statements) from the response, potentially leading to misjudgments in subsequent evidence collection and the final verdict.

**Questions:**

(1) Can the authors provide a detailed comparison of CLEAR (at \delta=0.8) with various baseline methods (such as GPT Labeling, GPT-ReCheck) in terms of average token consumption, API call frequency, or end-to-end latency?

(2) Could the design of the Extractor Agent lead to the loss of critical information? For instance, in a response like “I can't help you... but here’s a relevant general principle...”, might the extractor discard the rejection in the first half of the sentence, leading the Harm Score and Knowledge Judge agents to make incorrect judgments based on incomplete context? Have the authors conducted any ablation studies to assess the potential negative impact of this extractor?

(3)Could the authors elaborate on the advantages of this courtroom discussion approach over ranking methods like Elo and Hodge Rank?

---

### Official Review · Reviewer_rJio · 2025-11-01

**Soundness:** 1
**Presentation:** 2
**Contribution:** 1
**Rating:** 2
**Confidence:** 3

**Summary:**

The paper proposes CLEAR, a multi-agent jailbreak evaluation framework that simulates a courtroom process: multiple specialized LLM agents extract “evidence” about a model’s response (violation retrieval, intent fulfillment, harm score, and behavioral trajectory), a first-instance judge issues an initial verdict with confidence, and if the confidence is low, a second-stage debate between prosecution and defense agents refines the judgment. The authors claim this leads to more systematic and interpretable jailbreak evaluation and report results on three jailbreak datasets which they paired with model responses from GPT-3.5-turbo.

**Strengths:**

1. The paper identifies some issues in jailbreak evaluation (inconsistent metrics, lack of interpretability) and attempts to improve the consistency of the evaluations through a more sophisticated reasoning pipeline.

2. The framework is modular by design and could support different evaluators or safety policies, which gives it potential extensibility for future agent-based evaluation settings.

**Weaknesses:**

1. The method is only evaluated on responses provided by GPT-3.5-turbo, which is an obsolete model and can doubtfully provide harmful responses that are informative even when complying with jailbreaks. The effectiveness of the method for evaluating high-utility harmful responses produced by more recent and advanced models remains unknown.

2. The experiments largely focus on model responses and labels created by the authors themselves, with no evaluations on existing benchmarks (e.g. many datasets such as X-Teaming provide jailbreak prompts paired with responses). This raises strong concerns about overfitting the evaluation to a self-defined taxonomy.

3. The proposed framework is highly complex, which involves multiple stages and LLM components, making the stability and reproducibility of the evaluation results across different runs and conditions questionable. However, the paper does not provide any results to demonstrate evaluation consistency or robustness.

4. The paper lacks cost or latency analysis despite introducing 5+ agents and multi-round debate; it is unclear whether CLEAR is scalable beyond small curated datasets.

5. The supposed “court analogy” is mostly narrative packaging; the underlying pipeline does not introduce new technical contributions and appears to be an overengineered voting-plus-debate ensemble of LLM prompts.

**Questions:**

See above.

---

### Note · Authors · 2025-11-25

I have read and agree with the venue's withdrawal policy on behalf of myself and my co-authors.